# How Important is Importance Sampling for Deep Budgeted Training?

## Abstract

Long iterative training processes for Deep Neural Networks (DNNs) are commonly required to achieve state-of-the-art performance in many computer vision tasks. Core-set selection and importance sampling approaches might play a key role in budgeted training regimes, i.e. when limiting the number of training iterations. The former demonstrate that retaining informative samples is important to avoid large drops in accuracy, and the later aim at dynamically estimating the sample importance to speed-up convergence. This work explores this paradigm and how a budget constraint interacts with importance sampling approaches and data augmentation techniques. We show that under budget restrictions, importance sampling approaches do not provide a consistent improvement over uniform sampling. We suggest that, given a specific budget, the best course of action is to disregard the importance and introduce adequate data augmentation. For example, training in CIFAR-10/100 with 30% of the full training budget, a uniform sampling strategy with certain data augmentation surpasses the performance of 100% budget models trained with standard data augmentation. We conclude from our work that DNNs under budget restrictions benefit greatly from variety in the samples and that finding the right samples to train is not the most effective strategy when balancing high performance with low computational requirements. The code will be released after the review process.

## 1 Introduction

The availability of vast amounts of labeled data is crucial in training deep neural networks (DNNs) (Mahajan et al., 2018; Xie et al., 2020). Despite prompting considerable advances in many computer vision tasks (Yao et al., 2018; Sun et al., 2019a), this dependence poses two challenges: the generation of the datasets and the large computation requirements that arise as a result. Research addressing the former has experienced great progress in recent years via novel techniques that reduce the strong supervision required to achieve top results (Tan & Le, 2019; Touvron et al., 2019) by, e.g. improving semi-supervised learning (Berthelot et al., 2019; Arazo et al., 2020), few-shot learning (Zhang et al., 2018b; Sun et al., 2019b), self-supervised learning (He et al., 2020; Misra & Maaten, 2020), or training with noisy web labels (Arazo et al., 2019; Li et al., 2020a). The latter challenge has also experienced many advances from the side of network efficiency via DNN compression (Dai et al., 2018; Lin et al., 2019) or, neural architecture search (Tan & Le, 2019; Cai et al., 2019); and optimization efficiency by better exploiting the embedding space (Khosla et al., 2020; Kim et al., 2020). All these approaches are designed under a common constraint: the large dataset size needed to achieve top results (Xie et al., 2020), which conditions the success of the training process on computational resources. Conversely, a smart reduction of the amount of samples used during training can alleviate this constraint (Katharopoulos & Fleuret, 2018; Mirzasoleiman et al., 2020).

The selection of samples plays an important role in the optimization of DNN parameters during training, where Stochastic Gradient Descent (SGD) (Dean et al., 2012; Bottou et al., 2018) is often used. SGD guides the parameter updates using the estimation of model error gradients over sets of samples (mini-batches) that are uniformly randomly selected in an iterative fashion. This strategy assumes equal importance across samples, whereas other works suggest that alternative strategies for revisiting samples are more effective in achieving better performance (Chang et al., 2017; Kawaguchi & Lu, 2020) and faster convergence (Katharopoulos & Fleuret, 2018; Jiang et al., 2019). Similarly, the

selection of a unique and informative subset of samples (core-set) (Toneva et al., 2018; Coleman et al., 2020) can alleviate the computation requirements during training, while reducing the performance drop with respect to training on all data. However, while removing data samples speeds-up the training, a precise sample selection often requires a pretraining stage that hinders the ability to reduce computation (Mirzasoleiman et al., 2020; Sener & Savarese, 2018).

A possible solution to this limitation might be to dynamically change the important subset during training as done by importance sampling methods (Amiri et al., 2017; Zhang et al., 2019b), which select the samples based on a sampling probability distribution that evolves with the model and often changes based on the loss or network logits (Loshchilov & Hutter, 2015; Johnson & Guestrin, 2018). An up-to-date importance estimation is key for current methods to succeed but, in practice, is infeasible to compute (Katharopoulos & Fleuret, 2018). The real importance of a sample changes after every iteration and estimations become out-dated, yielding considerable drops in performance (Chang et al., 2017; Zhang et al., 2019b). Importance sampling methods, then, focus on selecting samples and achieve a speed-up during training as a side effect. They do not, however, strictly study possible benefits on DNN training when restricting the number of iterations used for training, i.e. the budget.

Budgeted training (Nan & Saligrama, 2017; Kachuee et al., 2019; Li et al., 2020b) imposes an additional constraint on the optimization of a DNN: a maximum number of iterations. Defining this budget provides a concise notion of the limited training resources. Li et al. (2020b) propose to address the budget limitation using specific learning rate schedules that better suit this scenario. Despite the standardized scenario that budgeted training poses to evaluate methods when reducing the computation requirements, there are few works to date in this direction (Li et al., 2020b; Katharopoulos & Fleuret, 2018). As mentioned, importance sampling methods are closely related, but the avoidance of budget restrictions makes it difficult to understand their utility given the sensitivity to hyperparameters that they often exhibit (Chang et al., 2017; Loshchilov & Hutter, 2015).

In this paper, we overcome the limitations outlined above by analyzing the effectiveness of importance sampling methods when a budget restriction is imposed (Li et al., 2020b). Given a budget restriction, we study synergies among important sampling, and data augmentation (Takahashi et al., 2018; Cubuk et al., 2020; Zhang et al., 2018a). We find the improvements of importance sampling approaches over uniform random sampling are not always consistent across budgets and datasets. We argue and experimentally confirm (see Section 4.4) that when using certain data augmentation (Takahashi et al., 2018; Cubuk et al., 2020; Zhang et al., 2018a), existing importance sampling techniques do not provide further benefits, making data augmentation the most effective strategy to exploit a given budget.

## 2 RELATED WORK

Few works exploit a budgeted training paradigm (Li et al., 2020b). Instead, many approaches aim to speed up the training convergence to a given performance by computing a better sampling strategy or carefully organizing the samples to allow the CNN to learn faster and generalize better. Other works, however, explore how to improve model performance by labeling the most important samples from an unlabeled set of data (Yoo & Kweon, 2019; Ash et al., 2020; Ren et al., 2020) or how to better train DNNs when a limited number of samples per class is available (Chen et al., 2019; Zhou et al., 2020; Albert et al., 2020). This section reviews relevant works aiming to improve the efficiency of the DNN training.

**Self-paced learning (SPL) and curriculum learning (CL)** aim to optimize the training process and improve model performance by ordering the samples from easy to difficult (Weinshall et al., 2018; Bengio et al., 2009; Hacohen & Weinshall, 2019; Cheng et al., 2019). For instance, CL manages to speed the convergence of the training at the initial stages due to focusing on samples whose gradients are better estimations of the real gradient (Weinshall et al., 2018). The main drawback of these methods is that, in most of the cases, the order of the samples (curriculum) has to be defined before training, which is already a costly task that requires manually assessing the sample difficulty, transferring knowledge from a fully trained model, or pre-training the model on the given dataset. Some approaches remedy this drawback with a simple curriculum (Lin et al., 2017) or by learning the curriculum during the training (Jiang et al., 2018); these methods, however, do not aim to speed up

the training by ordering the samples, but to improve network convergence by weighting the sample contribution to the loss.

**Core-set selection approaches** aim to find the subset of samples that is most useful (Toneva et al., 2018; Coleman et al., 2020; Mirzasoleiman et al., 2020). By identifying the most useful samples from a dataset, these methods aim at maintaining accuracy despite training in a subset of the data. The ability of these methods to reduce the training cost is very limited, since they require pre-training the model. However, these methods demonstrate that DNNs only need a portion of the samples to achieve peak performance. For example, Toneva et al. (2018) define "forgetting events" as the count of times that samples are miss-classified after being correctly predicted during training. They show that higher forgetting and importance are related, as removing samples with lower forgetting events damages the model less than removing the more forgotten ones. Mirzasoleiman et al. (2020) build clusters with the features from the model and use the centroids as the most informative samples. Coleman et al. (2020) demonstrate that the difficulty of a sample is invariant to the model capacity and show that they can speed up several sample selection tasks by reducing the size of the model.

**Importance sampling** approaches lie in the middle ground between the previous two: they aim to speed up training convergence by leveraging the most useful samples at every training stage (Katharopoulos & Fleuret, 2018; Jiang et al., 2019; Zhang et al., 2019b) – which correspond to sample losses with highest gradient magnitude (Needell et al., 2014; Zhao & Zhang, 2015; Alain et al., 2016). More recently, Johnson & Guestrin (2018) has shown that the last layer gradients are a good approximation and are easier to obtain in deep learning frameworks. Alternative importance measures often used include the loss (Jiang et al., 2019), the probability predicted for the true class (Chang et al., 2017), or the ranking order of these probabilities (Loshchilov & Hutter, 2015).

The approximation of the optimal distribution by importance sampling approaches avoids the cost of computing each sample importance at every iteration. However, they face one main challenge: the optimal sampling distribution changes very rapidly between iterations, leading to outdated estimations. Initial attempts on addressing this challenge included several hyper-parameters to smooth the estimated distribution (Chang et al., 2017), more frequent distribution updates via additional forward passes (Loshchilov & Hutter, 2015), or different alternative measures to estimate the sampling distribution (Amiri et al., 2017). Several works added complex support techniques to the training that aimed to estimate a better distribution: using robust optimization (Johnson & Guestrin, 2018), introducing repulsive point techniques (Zhang et al., 2019a), or adding a second network to be trained in parallel with the main model Zhang et al. (2019b). More recent methods leverage the random-then-greedy technique (Lu & Mazumder, 2018), where a random initial batch of samples is selected and then the probabilities of those samples are computed and used to select a secondary batch that is used for training. Within this scheme, (Katharopoulos & Fleuret, 2018) define a theoretical bound for the magnitude of the gradients that allows for faster computation of the sampling probabilities and (Jiang et al., 2019) and (Ioannou et al., 2019) use the loss as a measure of sample importance to keep the sampling distribution updated through the training. Finally, (Kawaguchi & Lu, 2020) introduces the top-$k$ loss from (Fan et al., 2017) to perform the back-propagation step using the samples with highest losses only. Note that none of these methods avoids doing a full forward pass every epoch to update the sampling probabilities.

**Learning rate schedules** have proven to be useful alternatives for faster convergence. The authors in (Smith & Topin, 2019; Smith, 2017) propose a cyclic learning rate schedule to reach faster convergence by using larger learning rates at intermediate training stages and very low rates at the end. Li et al. (Li et al., 2020b) also study the importance of the learning rate schedules to accelerate the training of DNNs. In particular, they explore budgeted training and propose a linearly decaying learning rate schedule that approaches zero at the end of the training, which without additional hyper-parameters, improves the standard learning rate schedules.

**Data augmentation** techniques, generally, aim to increase the variance of the data to achieve better generalization. Recent approaches, however, go a step further and target specific weaknesses from CNNs: cutout (DeVries & Taylor, 2017) drops contiguous patches of data from the input to force the network to spread its attention over the entire object, mixup (Zhang et al., 2018a) proposes to train using convex combinations of images and label pairs which smooth class boundaries and improve model calibration (Thulasidasan et al., 2019), and RICAP (Takahashi et al., 2018) combines the

advantages of the two previous techniques by training on images generated from joining multiple patches and doing the corresponding convex combination of labels. More generally, RandAugment (Cubuk et al., 2020) randomly combines commonly used data augmentation techniques as a reduction of the search space of the recently proposed methods that find automated augmentation policies (Ho et al., 2019; Cubuk et al., 2019).

## 3 BUDGETED TRAINING

The standard way of training DNNs is by gradient based minimization of cross-entropy

$$\ell(\theta) = -\frac{1}{N} \sum_{i=1}^{N} y_i^T \log h_\theta(y|x_i), \tag{1}$$

where $N$ is the number of samples in the dataset $D = \{x_i, y_i\}_{i=1}^{N}$ and $y_i \in \{0, 1\}^C$ is the one-hot encoding ground-truth label for sample $x_i$, $C$ is the number of classes, $h_\theta(y|x_i)$ is the predicted posterior probability of a DNN model given $x_i$ (i.e. prediction after a softmax normalization), and $\theta$ are the parameters of the model. Convergence to a reasonable performance usually determines the end of the training, whereas in budgeted training there is a fixed iteration budget. We adopt the setting by Li et al. (2020b), where the budget is defined as a percentage of the full training setup. Formally, we define the budget $B \in [0, 1]$ as the fraction of forward and backward passes used for training the model $h_\theta(x)$ with respect to a standard full training. As we aim at analyzing importance sampling, the budget restriction will be mainly applied to the amount of data $N \times B$ shown every epoch. However, a reduction on the number of epochs $T$ to $T \times B$ (where an epoch $T$ is considered a pass over all samples) is also considered as truncated training for budgeted training.

**Truncated training**   is the simplest approach to budgeted training: keep the standard SGD optimization and reduce the number of epochs trained by the model to $T \times B$. We call this strategy, where the model sees all the samples every epoch, *scan-SGD*. While seeing all the samples is common practice, we remove this constraint and draw the samples from a uniform probability distribution at every iteration and call this strategy *unif-SGD*. In this approach the budget is defined by randomly selecting $N \times B$ samples every epoch (and still training for $T$ epochs).

**Importance sampling**   aims to accelerate the convergence of SGD by sampling the most difficult samples $D_S = \{x_i, y_i\}_{i=1}^{N_S}$ more often, where $N_S = N \times B$ (the number of samples selected given a certain budget). Loshchilov & Hutter (2015) proposed a simple approach for importance sampling that uses the loss of every sample as a measure of the sample importance. Chang et al. (2017) adapts this approach to avoid additional forward passes by using as importance:

$$p_i^t = \frac{1}{t} \sum_{k=1}^{t} \left( 1 - y_i^T h_\theta^k(y|x_i) \right) + \epsilon^t, \tag{2}$$

where $h_\theta^k(y|x_i)$ is the prediction of the model given the sample $x_i$ in epoch $k$, and $t$ is the current epoch. Therefore, the average predicted probability across previous epochs associated to the ground-truth class of each sample defines the importance of sample $x_i$. The smoothing constant $\epsilon^t$ is defined as the mean per sample importance up to the current epoch: $\frac{1}{N} \sum_{i=1}^{N} p_i^t$.

The sampling distribution $P^t$ at a particular epoch $t$ is then given by:

$$P_i^t = \frac{p_i^t}{\sum_{j=1}^{N} p_j}. \tag{3}$$

By drawing samples from the distribution $P^t$ this approach biases the training towards the most difficult samples, and selects those samples with highest loss value; we name this method *p-SGD*. Similarly, (Chang et al., 2017) propose to select those samples that are closer to the decision boundaries and favors the samples with higher uncertainty by defining the importance measure as $c_i^t = p_i^t \times (1 - p_i^t)$; we name this approach *c-SGD*.

Both *p-SGD* and *c-SGD* are very computationally efficient as the importance estimation only requires information available during training. Conversely, Jiang et al. (2019) propose to perform forward

passes on all the samples to determine the most important ones and later reduce the amount of backward passes; they name this method selective backpropagation (*SB*). At every forward pass, *SB* stores the sample $x_i$ with probability:

$$s_i^t = \left[ F_R(\ell(h_\theta^t(x_i), y_i)) \right]^b, \tag{4}$$

where $F_R$ is the cumulative distribution function from a history of the last $R$ samples seen by the model and $b > 0$ is a constant that determines the selectivity of the method, i.e. the budget used during the training. In practice, *SB* does as many forward passes as needed until it has enough samples to form a full a mini-batch. It then performs the training forward and backward passes with the selected samples to update the model.

Finally, as an alternative training paradigm to prioritize the most important samples, Kawaguchi & Lu (2020) propose to use only the $q$ samples with highest loss from a mini-batch in the backward pass. As the training accuracy increases, $q$ decreases until only $1/16$ th of the images in the mini-batch are used in the backward pass. The authors name this approach *ordered* SGD (*OSGD*) and provide a default setting for the adaptive values of $q$ depending on the training accuracy.

**Importance sampling methods under budgeted training**  give a precise notion of the training budget. For *unif-SGD*, *p-SGD*, and *c-SGD* the adaptation needed consists of selecting a fixed number of samples per epoch $N \times B$ based on the corresponding sampling probability distribution $P_t$ and still train the full $T$ epochs. For *SB*, the parameter $b$ determines the selectivity of the algorithm: higher values will reject more samples. Note that this method requires additional forward passes that we exclude from the budget as they do not induce the backward passes used for training. We adapt *OSGD* by truncating the training as in *scan-SGD*: all the parameters are kept constant but the total number of epochs is reduced to $T \times B$. Additionally, we consider the wall-clock time of each method with respect to a full budget training as a metric to evaluate the approaches.

## 4 EXPERIMENTS AND RESULTS

### 4.1 EXPERIMENTAL FRAMEWORK

**Datasets**  We experiment on image classification tasks using CIFAR-10/100 (Krizhevsky et al., 2009), SVHN (Netzer et al., 2011), and mini-ImageNet (Vinyals et al., 2016) datasets. CIFAR-10/100 consist of 50K samples for training and 10K for testing; each divided into 10(100) classes for CIFAR-10(100). The samples are images extracted from ImageNet (Deng et al., 2009) and down-sampled to 32×32. SVHN contains 32×32 RGB images of real-world house numbers divided into 10 classes, 73257 for training and 26032 for testing. Mini-ImageNet is a subset of ImageNet with 50K samples for training and 10K for testing divided into 10 classes and down-sampled to 84×84. Unless otherwise stated, all the experiments use standard data augmentation: random cropping with padding of 4 pixels per side and random horizontal flip.

**Training details**  We train a ResNet-18 architecture (He et al., 2016) for 200 epochs with SGD with momentum of 0.9 and a batch size of 128. We use two learning rate schedules: step-wise and linear decay. For both schedules we adopt the budget-aware version proposed by Li et al. (2020b) and use an initial learning rate of 0.1. In the step-wise case, the learning rate is divided by 10 at 1/3 (epoch 66) and 2/3 (epoch 133) of the training. The linear schedule decreases the learning rate value at every iteration linearly from the initial value to approximately zero ($10^{-6}$) at the end of the training. We always report the average accuracy and standard deviation of the model across 3 independent runs trained on a GeForce GTX 1080Ti GPU.

### 4.2 BUDGET-FREE TRAINING FOR IMPORTANCE SAMPLING

Current methods from the state-of-the-art are optimized with no restriction in the number of training iterations. While this allows the methods to better exploit the training process, it makes it difficult to evaluate their computational benefit. Therefore, Table 1 presents the performance, wall-clock time, and speed-up relative to a full training of the methods presented in Section 3. While the simpler approaches to importance sampling, *p-SGD* and *c-SGD*, achieve similar performance to SGD and

Table 1: Test accuracy, time and speed-up (reduction with respect SGD) in CIFAR-10/100 under a budget-free training. (*) denotes that we have used the official code.

| Method | CIFAR-10 | | | CIFAR-100 | | |
|---|---|---|---|---|---|---|
| | Accuracy (%) | Time (min) | Speed-up (%) | Acuracy | Time | Speed-up |
| SGD | $94.58 \pm 0.33$ | 141 | 0.00% | $74.56 \pm 0.06$ | 141 | 0.00% |
| *p-SGD* | $94.41 \pm 0.19$ | 113 | 19.86% | $74.44 \pm 0.06$ | 127 | 9.93% |
| *c-SGD* | $94.17 \pm 0.11$ | 100 | 29.08% | $74.40 \pm 0.06$ | 127 | 9.93% |
| *SB* (*) | $93.90 \pm 0.16$ | 85 | 39.72% | $73.39 \pm 0.37$ | 119 | 15.60% |
| *OSGD* (*) | $94.34 \pm 0.07$ | 139 | 0.07% | $74.22 \pm 0.21$ | 141 | 0.00% |

reduce the computational time up to 29.08 % (9.93%) in CIFAR-10 (CIFAR-100), *SB* reduces the training time 39.72% (15.60%) in CIFAR-10 (CIFAR-100) with very small drops in accuracy.

All methods train with a step-wise linear learning rate schedule. SGD corresponds to a standard training as described in Subsection 4.1. *p-SGD* and *c-SGD* correspond to the methods described in Section 3 introduced by (Chang et al., 2017) that for the experiments in Table 1 train for 200 epochs where the first 70 epochs consist of a warm-up stage with a uniform sampling strategy as done in the original paper. For CIFAR-10 we use a budget of 0.8 for *p-SGD* and 0.7 for *c-SGD*, and for CIFAR-100 a budget of 0.9 for both approaches (budgets retaining most accuracy were selected). Finally, *SB* and *OSGD* follow the setups described in the corresponding papers, (Jiang et al., 2019) and (Kawaguchi & Lu, 2020), and run on the official code.

### 4.3 BUDGETED TRAINING FOR IMPORTANCE SAMPLING

We adapt importance sampling approaches as described in Section 3 and configure each method to constrain its computation to the given budget. Table 2 shows the analyzed methods performance under the same budget for a step-wise learning rate (SLR) decay and the linear decay (LLR) proposed by Li et al. (2020b) for budgeted training (described in Section 4.1). Surprisingly, this setup shows that most methods achieve very similar performance given a predefined budget, thus not observing faster convergence when using importance sampling. Both *p-SGD* and *c-SGD* provide marginal or no improvements: *p-SGD* improves *unif-SGD* in CIFAR-10 with a step-wise schedule of the learning rate, but fails to do so in CIFAR-100, and in the LLR setup only improves for certain budgets. Similar behaviour is observed in the results from *c-SGD*. Conversely, *SB* surpasses the other approaches consistently for SLR and in most cases in the LLR setup. However, *SB* introduces additional forward passes not considered as budget, while the other methods do not.

We consider *scan-SGD* and *unif-SGD*, as two naive baselines for budgeted training. Despite having similar results (*scan-SGD* seems to be marginally better than *unif-SGD*), we use *unif-SGD* for further experimentation in the following subsections as it adopts a uniform random sampling distribution, which contrast alternative sampling distributions of importance sampling methods.

Additionally, Table 2 confirms the effectiveness of a linear learning rate schedule as proposed in (Li et al., 2020b): all methods consistently improve with this schedule and in most of the cases *unif-SGD* and LLR performs on par with *SB* and SLR and surpasses all the other methods when using SLR.

### 4.4 DATA VARIABILITY IMPORTANCE DURING TRAINING

Core-set selection approaches (Toneva et al., 2018; Coleman et al., 2020) aim to find the most representative samples in the dataset to make training more efficient, while keeping accuracy as high as possible. Figure 1 (top) presents how core-set selection and a randomly chosen subset (Random) both under-perform randomly sampling from a uniform distribution a different subset every epoch (*unif-SGD*), which approaches a standard training (black dashed line). Therefore, this experiment shows that varying the important subset during training (*unif-SGD*) is equally efficient from a training computation perspective, while bringing substantially better accuracy. Moreover, we find data variability to play an important role within importance sampling. We report our experiments comparing data variability in Figure 1 (bottom), where data variability is measured using the entropy $H(c)$ of the number of times that a sample is presented to the network during training, being $c$ the normalized $N$-dimension vector with the counts of each sample. Figure 1 (bottom) shows how

Table 2: Test accuracy with a step-wise and a linear learning rate decay under different budgets. Note that *SB* requires additional computation (forward passes).

| | CIFAR-10 | | | CIFAR-100 | | |
|---|---|---|---|---|---|---|
| *SGD - SLR* | $94.58 \pm 0.33$ | | | $74.56 \pm 0.06$ | | |
| *SGD - LLR* | $94.80 \pm 0.08$ | | | $75.44 \pm 0.16$ | | |
| Budget: | 0.2 | 0.3 | 0.5 | 0.2 | 0.3 | 0.5 |
| | Step-wise decay of the learning rate (SLR) | | | | | |
| *scan-SGD* | $92.03 \pm 0.24$ | $93.06 \pm 0.15$ | $93.80 \pm 0.15$ | $70.89 \pm 0.23$ | $72.31 \pm 0.22$ | $73.49 \pm 0.20$ |
| *unif-SGD* | $92.04 \pm 0.14$ | $92.86 \pm 0.25$ | $93.80 \pm 0.21$ | $70.46 \pm 0.39$ | $71.71 \pm 0.05$ | $73.23 \pm 0.47$ |
| *p-SGD* | $92.28 \pm 0.05$ | $92.91 \pm 0.18$ | $93.85 \pm 0.07$ | $70.24 \pm 0.28$ | $72.11 \pm 0.39$ | $72.94 \pm 0.36$ |
| *c-SGD* | $91.70 \pm 0.25$ | $92.83 \pm 0.30$ | $93.71 \pm 0.15$ | $69.86 \pm 0.36$ | $71.56 \pm 0.27$ | $73.02 \pm 0.34$ |
| *SB* | $93.37 \pm 0.11$ | $93.86 \pm 0.27$ | $94.21 \pm 0.13$ | $70.94 \pm 0.38$ | $72.25 \pm 0.68$ | $73.39 \pm 0.37$ |
| *OSGD* | $90.61 \pm 0.31$ | $91.78 \pm 0.30$ | $93.45 \pm 0.10$ | $70.09 \pm 0.25$ | $72.18 \pm 0.35$ | $73.39 \pm 0.22$ |
| | Linear decay of the learning rate (LLR) | | | | | |
| *scan-SGD* | $92.95 \pm 0.07$ | $93.55 \pm 0.21$ | $94.22 \pm 0.16$ | $\mathbf{72.04 \pm 0.42}$ | $72.97 \pm 0.07$ | $73.90 \pm 0.43$ |
| *unif-SGD* | $92.94 \pm 0.19$ | $93.66 \pm 0.16$ | $94.19 \pm 0.12$ | $71.71 \pm 0.11$ | $72.59 \pm 0.14$ | $73.99 \pm 0.27$ |
| *p-SGD* | $93.23 \pm 0.14$ | $93.63 \pm 0.04$ | $94.14 \pm 0.11$ | $71.72 \pm 0.37$ | $72.94 \pm 0.37$ | $74.06 \pm 0.10$ |
| *c-SGD* | $92.95 \pm 0.17$ | $93.54 \pm 0.07$ | $94.11 \pm 0.24$ | $71.37 \pm 0.49$ | $72.33 \pm 0.18$ | $73.93 \pm 0.35$ |
| *SB* | $\mathbf{93.78 \pm 0.11}$ | $\mathbf{94.06 \pm 0.37}$ | $\mathbf{94.57 \pm 0.18}$ | $71.96 \pm 0.67$ | $\mathbf{73.11 \pm 0.42}$ | $\mathbf{74.35 \pm 0.34}$ |
| *OSGD* | $91.87 \pm 0.36$ | $93.00 \pm 0.08$ | $93.93 \pm 0.22$ | $71.25 \pm 0.11$ | $72.56 \pm 0.36$ | $73.40 \pm 0.14$ |

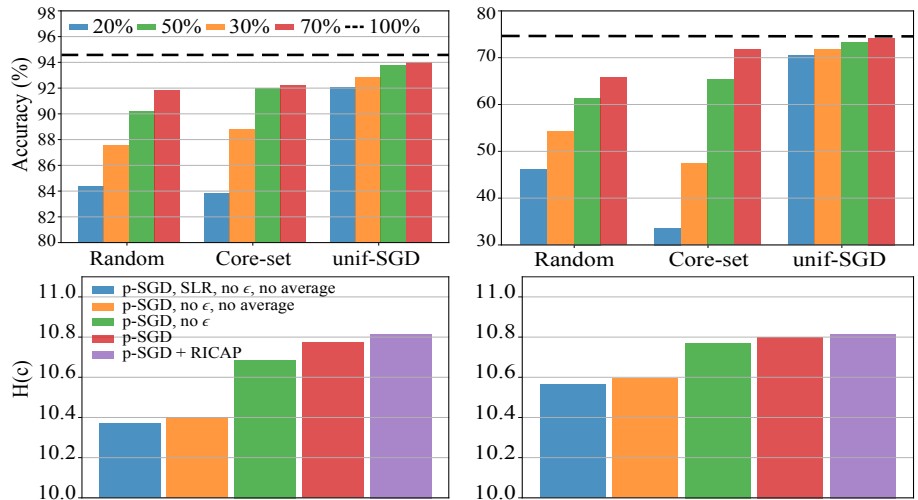

Figure 1: Importance of data variability in CIFAR-10 (left) and CIFAR-100 (right). Top: randomly selecting samples at every epoch (*unif-SGD*) outperforms fixed core-set or random subsets. Bottom: the entropy of sample counts during training (0.3 budget) demonstrates that importance sampling, linear learning rate, and data augmentation contribute to higher data variability (entropy).

improvements in *p-SGD* relate to higher data variability (higher entropy): adding to the P sampling distribution from *p-SGD* the LLR, the smoothing constant, the average of the predictions across epochs, and data augmentation.

## 4.5 DATA AUGMENTATION FOR IMPORTANCE SAMPLING

Importance sampling approaches usually do not explore the interaction of sampling strategies with data augmentation techniques (Loshchilov & Hutter, 2015; Katharopoulos & Fleuret, 2018; Jiang et al., 2019). To better understand this interaction, we explore interpolation-based augmentations via RICAP (Takahashi et al., 2018) and mixup (Zhang et al., 2018a); and non-interpolation augmentations using RandAugment (Cubuk et al., 2020). We implemented these data augmentation policies as reported in the original papers (see Table 3 for the hyperparameters used in our experiments). Note

Table 3: Data augmentation for budgeted importance sampling. N and M are the number and strength of RandAugment augmentations, and $\alpha$ controls the interpolation in mixup and RICAP. Note that SGD corresponds to the full training.

| | CIFAR-10 | | | CIFAR-100 | | |
|---|---|---|---|---|---|---|
| Budget: | 0.2 | 0.3 | 0.5 | 0.2 | 0.3 | 0.5 |
| | Standard data augmentation | | | | | |
| SGD ($B = 1$) | | $94.80 \pm 0.08$ | | | $75.44 \pm 0.16$ | |
| *unif-SGD* | $92.94 \pm 0.19$ | $93.66 \pm 0.16$ | $94.19 \pm 0.12$ | $71.71 \pm 0.11$ | $72.59 \pm 0.14$ | $73.99 \pm 0.27$ |
| *p-SGD* | $93.23 \pm 0.14$ | $93.63 \pm 0.04$ | $94.14 \pm 0.11$ | $71.72 \pm 0.37$ | $72.94 \pm 0.37$ | $74.06 \pm 0.10$ |
| *SB* | $93.78 \pm 0.11$ | $94.06 \pm 0.37$ | $94.57 \pm 0.18$ | $71.96 \pm 0.67$ | $73.11 \pm 0.42$ | $74.35 \pm 0.34$ |
| | RandAugment data augmentation (N = 2, M = 4) | | | | | |
| SGD ($B = 1$) | | $95.56 \pm 0.12$ | | | $75.52 \pm 0.17$ | |
| *unif-SGD* | $92.68 \pm 0.17$ | $94.06 \pm 0.15$ | $94.83 \pm 0.10$ | $71.31 \pm 0.35$ | $73.68 \pm 0.15$ | $74.61 \pm 0.23$ |
| *p-SGD* | $92.95 \pm 0.31$ | $93.99 \pm 0.28$ | $94.91 \pm 0.18$ | $71.63 \pm 0.27$ | $72.91 \pm 0.13$ | $74.30 \pm 0.04$ |
| *SB* | $93.27 \pm 0.38$ | $94.64 \pm 0.07$ | $95.27 \pm 0.26$ | $66.84 \pm 1.15$ | $73.79 \pm 0.40$ | $74.87 \pm 0.18$ |
| | mixup data augmentation ($\alpha = 0.3$) | | | | | |
| SGD ($B = 1$) | | $95.82 \pm 0.17$ | | | $77.62 \pm 0.40$ | |
| *unif-SGD* | $93.64 \pm 0.27$ | $94.49 \pm 0.04$ | $95.18 \pm 0.05$ | $73.28 \pm 0.51$ | $75.13 \pm 0.52$ | $75.80 \pm 0.34$ |
| *p-SGD* | $93.78 \pm 0.04$ | $94.41 \pm 0.16$ | $95.26 \pm 0.06$ | $73.35 \pm 0.29$ | $75.05 \pm 0.15$ | $75.87 \pm 0.15$ |
| *SB* | $93.62 \pm 0.36$ | $93.92 \pm 0.08$ | $94.51 \pm 0.17$ | $73.38 \pm 0.13$ | $74.88 \pm 0.31$ | $75.57 \pm 0.23$ |
| | RICAP data augmentation ($\alpha = 0.3$) | | | | | |
| SGD ($B = 1$) | | $96.17 \pm 0.09$ | | | $78.91 \pm 0.07$ | |
| *unif-SGD* | $94.00 \pm 0.19$ | $\mathbf{94.85 \pm 0.14}$ | $95.58 \pm 0.06$ | $\mathbf{74.86 \pm 0.10}$ | $\mathbf{76.65 \pm 0.05}$ | $\mathbf{77.74 \pm 0.17}$ |
| *p-SGD* | $\mathbf{94.02 \pm 0.18}$ | $94.79 \pm 0.18$ | $\mathbf{95.63 \pm 0.15}$ | $74.59 \pm 0.15$ | $76.50 \pm 0.22$ | $77.58 \pm 0.49$ |
| *SB* | $89.93 \pm 0.84$ | $93.64 \pm 0.42$ | $94.76 \pm 0.02$ | $56.66 \pm 0.65$ | $72.24 \pm 0.58$ | $76.26 \pm 0.22$ |

Table 4: Data augmentation for budgeted importance sampling in SVHN and mini-ImageNet. Note that SGD corresponds to the full training.

| | SVHN | | | mini-ImageNet |
|---|---|---|---|---|
| Budget: | 0.2 | 0.3 | 0.5 | 0.3 |
| | Standard data augmentation | | | |
| SGD ($B = 1$) | | $97.02 \pm 0.05$ | | $75.19 \pm 0.16$ |
| *unif-SGD* | $96.56 \pm 0.12$ | $96.78 \pm 0.13$ | $96.95 \pm 0.07$ | $72.19 \pm 0.43$ |
| *p-SGD* | $96.56 \pm 0.12$ | $96.77 \pm 0.01$ | $96.87 \pm 0.05$ | $72.39 \pm 0.45$ |
| *SB* | $96.93 \pm 0.07$ | $96.85 \pm 0.01$ | $96.97 \pm 0.06$ | $71.46 \pm 0.15$ |
| | RICAP data augmentation ($\alpha = 0.3$) | | | |
| SGD ($B = 1$) | | $97.61 \pm 0.06$ | | $78.75 \pm 0.40$ |
| *unif-SGD* | $97.47 \pm 0.04$ | $97.62 \pm 0.16$ | $97.55 \pm 0.04$ | $75.15 \pm 0.45$ |
| *p-SGD* | $97.48 \pm 0.08$ | $97.50 \pm 0.05$ | $97.51 \pm 0.08$ | $75.46 \pm 0.27$ |
| *SB* | $97.34 \pm 0.03$ | $97.40 \pm 0.06$ | $97.45 \pm 0.01$ | $71.75 \pm 0.67$ |

that in mixup and RICAP we combine 2 and 4 images respectively within each minibatch, which results in the same number of samples being shown to the network ($T \times B$).

Table 3 and 4 show that data augmentation is beneficial in a budgeted training scenario, in most cases all strategies studied increase performance of the different methods compared to the standard data augmentation. The main exception is in the lowest budget for *SB* where in some cases data augmentation damages performance. In particular, with RICAP and mixup, the improvements from importance sampling approaches are marginal and the naive *unif-SGD* provides results close to full training with standard data augmentation. In some cases *unif-SGD* surpasses full-training with standard augmentations, e.g. RICAP with 0.3 and 0.5 of budget in CIFAR-100, and both mixup and RICAP with 0.3 of budget in CIFAR-10. Note that this is even more evident in SVHN where all the budgets in Table 4 for *unif-SGD* with RICAP surpass the full training (SGD) with standard data augmentation.

Given the cost of the data augmentation policies considered is negligible (see Appendix B for details on wall-clock times), our results show that an adequate data augmentation can reduce the training time at no cost of accuracy and in some cases with a considerable increase in accuracy. For example, a 70% reduction of the training time (0.3 budget) corresponds to an increase in accuracy from 75.44% to 76.65% in CIFAR-100 and from 94.80% to 94.85% in CIFAR-10. Also, a 50% reduction (0.5 budget) corresponds to an increase in accuracy from 75.44% to 77.78% in CIFAR-100 and from 94.80% to 95.58% in CIFAR-10.

We also experimented with extremely low budgets (0.05 and 0.1) and found that data augmentation damages the training of DNNs (see Appendix A). For example, with $B = 0.05$ there is a drop of approximately 3 points in accuracy in CIFAR-10 and 5 points in CIFAR-100 with respect 88.34% and 62.84% for *unif-SGD* with standard data augmentation.

## 5 CONCLUSION

This paper studies DNN training when the number of iterations is fixed (i.e. budgeted training) and explores the interaction of importance sampling techniques and data augmentation in this setup. Our experimental results suggest that, in budgeted training, DNNs prefer variability over selection of important samples: adequate data augmentation surpasses state-of-the-art importance sampling methods and allows for up to a 70% reduction of the training time (budget) with no loss or even increase in accuracy. Given the strong impact that data augmentation has in improving performance of budgeted training, we consider as interesting future work, exploring the limitations found in extreme budgets and in extending the study to large-scale datasets where training DNNs becomes a long-lasting process. Finally, the results presented in this paper motivate research in the direction of exploring training techniques and methodologies to further exploit budgeted training.

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

Table 5: Test accuracy for CIFAR-10/100 under extreme budgets.

| | CIFAR-10 | | CIFAR-100 | |
|---|---|---|---|---|
| Budget: | 0.05 | 0.1 | 0.05 | 0.1 |
| Standard data augmentation | | | | |
| *unif-SGD* | $88.34 \pm 0.26$ | $91.19 \pm 0.11$ | $62.84 \pm 0.07$ | $69.25 \pm 0.51$ |
| *p-SGD* | $88.86 \pm 0.17$ | $91.66 \pm 0.11$ | $62.69 \pm 0.33$ | $69.62 \pm 0.50$ |
| *SB* | $79.45 \pm 4.31$ | $92.66 \pm 0.14$ | $50.53 \pm 2.27$ | $68.29 \pm 0.68$ |
| RICAP data augmentation ($\alpha = 0.3$) | | | | |
| *unif-SGD* | $85.54 \pm 0.40$ | $91.27 \pm 0.18$ | $57.55 \pm 0.60$ | $69.38 \pm 0.58$ |
| *p-SGD* | $85.57 \pm 0.70$ | $90.94 \pm 0.16$ | $56.09 \pm 0.71$ | $70.05 \pm 0.07$ |
| *SB* | $44.93 \pm 2.67$ | $54.76 \pm 4.31$ | $10.75 \pm 0.72$ | $13.33 \pm 0.39$ |

Table 6: Wall-clock time (minutes) in CIFAR-100 for a training of 0.3 of budget.

| Approaches: | *unif-SGD* | *p-SGD* | *SB* |
|---|---|---|---|
| Standard data augmentation | 47 | 48 | 91 |
| RICAP | 49 | 49 | 95 |

## A EXTREME BUDGETS

Table 5 shows the performance of the different approaches when the budget is further reduced to 0.05 and 0.1. These results show that in this extreme scenario, importance sampling approaches (*s-SGD* and *SB*) still bring little improvement over randomly selecting the training samples (*unif-SGD*). However, additional data augmentation does not bring a significant improvement in accuracy and in the most challenging cases, hinders convergence.

## B WALL-CLOCK TIME

Table 6 shows the wall-clock time in minutes corresponding to 0.3 of budget in CIFAR-100 for *unif-SGD*, *p-SGD*, and *SB* under different data augmentation policies. Note that *SB* has higher training times due to the additional forward passes introduced to compute the sample importance.

