# OpenReview forum: "How Important is Importance Sampling for Deep Budgeted Training?"
_ICLR.cc/2021/Conference — Reject_

### Official Review · AnonReviewer2 · 2020-10-26
**This paper studies how several importance sampling strategies jointly applied with data-augmentation technics impact the performances of deep budgeting training**

**Rating:** 4
**Confidence:** 2

**Review:**

This experimental contribution concludes that importance sampling strategies do not improve training with budgeted constraints. Data Augmentation seems to be a better strategy in this case.
The paper is easy to read and the experiments are clearly stated.
Data augmentation and sampling strategies and two different technics. Comparing them in a jointly way seems not natural. Figure 1 showing the variability of the data may be a commun link between these two methods since they have some effect on entropy.
Table 3 shows that the effect of Data augmentation  is close for both SGD model and Sampling strategies. Why SGD-based accuracy is not reported for budget = 0.2 an 0.5? It seems to be important to validate that data augmentation is beneficial for both SGD and Sampling technics.
Additional experiment should be achieved in largest and more challenging datasets (at least imagenet) to show if the conclusions are the same.

The budget aware version proposed by Li and Al should be detailed and as in the paper, it should be interesting to report some training/val loss vs epoch to compare the convergence of the tested strategies

---

> ### Author Response · Authors · 2020-11-18
> **Response to Reviewer 2**
>
> We thank the reviewer for the valuable feedback and for the suggestions that we believe will improve the paper.
>
> #### **Clarify the main point:**
> We agree with the reviewer in that combining these two strategies (data augmentation and importance sampling) seems unnatural. However, when training under budget restrictions we have found that DNNs converge to better accuracy when the variety of the samples is increased (through data augmentation) than when the training focuses on particular samples.
>
> The final version of the paper will also include experiments in other datasets (SVHN, and mini-ImageNet) to make the conclusions stronger (concern also raised by Rev. 1 and 4). Please, see “Further exploration” in the reply for Rev. 1 for a thorough explanation and preliminary results.
>
> #### **Additional comments:**
> The row corresponding to SGD in Table 3 reports results for 100% of the budget (B = 1), not for B = 0.3. We will clarify this in the final version of the paper.
>
> We will review the text to make sure that the budget-aware versions of the learning rate schedules are properly explained and  we will include the training/validation vs epochs curves in the appendix.

---

### Official Review · AnonReviewer4 · 2020-10-28
**Interesting comparison, that should be extended**

**Rating:** 4
**Confidence:** 4

**Review:**

# Summary
This paper studies the interplay of budgeted training, data augmentation, learning rate schedules and importance sampling and finds data augmentation to play a key role.

# Score and reasons
Overall I vote for rejecting, but encourage the authors to extend their study to more learning rates and more datasets, as well as provide more statistics.
While I appreciate the idea of comparing the sampling methods and learning rates in a budgeted training setting, such a study should be more comprehensive or suggest a new method as a conclusion.


# Strong / Weak points
## Pros
This paper aims to connect important research directions, i.e. given a specific training budget, i.e. epochs, how can the best performance be achieved.
The paper provides an interesting comparison of the speed/accuracy trade-off of several methods.

## Cons
The paper is merely a comparison of existing methods, which is as such interesting, a broader range of datasets would be beneficial.
Stating to study the influence of learning rate schedules, while using only two such schedules is not enough. The authors should enhance their study with further examples, e.g. cyclic learning rate [https://arxiv.org/pdf/1506.01186.pdf]
Clarity of the presentation could be improved.
E.g. introduction and related work are not clearly separated
Much of the paper is spend on explaining contributions by others, while remains elusive what the novel contribution of this paper is.

# Questions to the authors
Please provide more details on the data augmentation you used, e.g. how many samples are shown to the network during training, including the variations through data augmentation.

How does data augmentation compare in terms of wall-time?

# Detailed comments
Abstract could be more concise.
Introduction is missing a common leitmotif connecting the paragraphs and seems to anticipate some content that would better be placed in the related work section.

Related work
The statements on curriculum learning are not complete.
Not all curriculum methods require pre-training a model, a simple curriculum is the focal-loss Lin, T., Goyal, P., Girshick, R.B., He, K., & Dollár, P. (2020). Focal Loss for Dense Object Detection. IEEE Transactions on Pattern Analysis and Machine Intelligence, 42, 318-327. [http://arxiv.org/pdf/1708.02002]. Methods like MentorNet jointly learn the curriculum from data, Jiang, L., Zhou, Z., Leung, T., Li, L., & Fei-Fei, L. (2018). MentorNet: Learning Data-Driven Curriculum for Very Deep Neural Networks on Corrupted Labels. ICML. [http://proceedings.mlr.press/v80/jiang18c/jiang18c.pdf]

Methods for estimating the importance are also highly used in active learning, relating to that, e.g. margin sampling, ensemble variation etc. would be beneficial, for a survey of established methods see e.g. Settles, B. (2009). Active Learning Literature Survey. [http://axon.cs.byu.edu/~martinez/classes/778/Papers/settles.activelearning.pdf]

Section on learning rate schedules:  "have proven to be useful alternatives for faster convergence (Smith, 2017;
Smith & Topin, 2019; Li et al., 2020b). The authors in (Smith & Topin, 2019; Smith, 2017)" --> clutters the text to cite the very same papers twice in a row

Avoid vague language where possible, e.g. section 3: "DNN are usually trained" --> if you need that formula for further explanations state it as an assumption, if not, why have it at all? If they are only trained "usually" with a loss function, what are the alternatives?
DNN is missing an "s" here

Early stopping is probably not an optimal name, as this usually refers to the regularization method of "early-stopping", meaning terminating a training when the validation accuracy drops.

Deriving statistics from three runs seems to be not very informative, I would suggest to use a least 5 runs.

Highlighting the best results in your tables could guide the reader.

---

> ### Author Response · Authors · 2020-11-18
> **Response to Reviewer 4**
>
> We thank the reviewer for the valuable feedback and for the suggestions that we believe will improve the paper.
>
> #### **Further exploration:**
> Li et al. (Li et al., 2020b) provides a comprehensive study on the interactions between learning rate schedules and budget training. Our paper builds upon the research findings from  Li et al. and adopts their framework to further study the training of DNNs under budget restrictions. We will amend the text to make it clear that we do not aim to study the interaction between learning rate and budgeted training or importance sampling.
>
> We agree with the reviewer in that more datasets would give a more robust picture (concern also raised by Rev. 1 and 2), we will address this point in the final version of the paper. Please, see “Further exploration” in the reply for Rev. 1 for a thorough explanation and preliminary results.
>
> #### **Clarify the main point:**
> We will address the concerns regarding the introduction and related work sections (also raised by Rev. 3) and move the discussion about related work from the introduction to the related work and condense the text to transmit the main point of the paper more clearly: to reduce the drop in accuracy when decreasing the training budget, it is better to introduce data augmentation rather than selecting specific samples.
>
> #### **Question to authors:**
> The number of samples shown to the network through training depends on the budget regardless of the data augmentation: the model sees B x N, where B is the budget and N is the number of samples in the dataset. The implementations that we follow for the different data augmentation are the ones reported in the original papers (we combine 2 images in mixup and 4 in RICAP, note that the images are mixed within a minibatch). We will clarify this in the text.
>
> We forgot to mention that the increase of wall-clock time introduced by the data augmentation policies that we considered is negligible, we will update the text to mention this and include a table with wall-clock times in the appendix of the paper.
>
> #### **Additional citations:**
> We thank the reviewer for the suggestion about the focal loss and MentorNet, we will include them in the related work and amend the text to avoid the assumption that curriculum learning methods always require pre-training. We will also include a reference to active learning in the related work because as the reviewer mentions, they use importance measures to select samples to be labeled.
>
> #### **Writing suggestions:**
> We will update the manuscript and incorporate the following changes in the text:
> - Remove repeated citations in the learning rate section in literature review
> - Rephrase the beginning of Section 3.
> - Highlight the best results in the tables
>
> We thank the reviewer for noticing that “Early stopping” is a confusing name. While this approach is the same as early stopping, the purpose is different. Hence, we will rename it as “Truncated training”.
>
> Finally, we will update the results of the paper with 5 runs per experiment before the end of the review process.

---

> > ### Comment · AnonReviewer4 · 2020-11-25
> > **Post-rebuttal**
> >
> > I would like to thank the authors for their additional explanations.
> > As I still feel some insights and connection missing between (the very valid) points raised in the paper I'll stick to my previous score.
> >
> > Some remaining points include e.g.:
> > Regarding data augmentation the amount of total steps performed in the training e.g. as per
> > steps_per_epoch = train_size // batch_size
> > should be relevant here. It is the total amount of optimization steps performed in the training that needs to be considered. E.g. if you would restrict to $B\cdot N$ samples and increase the number through data augmentation (A-fold, where $A\in \mathbb{N}_+$) to $A\cdot B\cdot N$, you would be comparing different budgets.
> > As for the wall-times provided, you used the "official code" for some results, can it be implied, that all methods that are compared in their wall-time use the same DL framework?

---

### Official Review · AnonReviewer3 · 2020-10-29
**Reviewer #3**

**Rating:** 3
**Confidence:** 4

**Review:**

Overview:

The paper studies the effect of important sampling approaches in the context of budgeted training. They empirically show that important sampling does not provide consistent improvement over uniform sampling. Instead they find that the budgeted training benefits from variety in the sampled introduced by data augmentation.


Strengths:

++ Both the average accuracy and the standard deviation are reported across 3 runs. This makes the experiment results more statically convincing.

++ The literature survey on budgeted training as well as importance sampling is detailed and clear.


Weaknesses:

-- Inappropriate title. The paper argues that " under budge restrictions, importance sampling approaches do NOT provide a consistent improvement ..." and the useful part is the data augmentation. Then why the title is the importance of importance sampling (instead of data augmentation)?

-- The motivation of this paper is unclear.  The majority of the introduction looks like a duplicate related work to me.

-- The paper is not well organized. For example, the reason for including core-set selection is unclear. In Sec 4, the authors conduct experiments by adapting importance sampling approaches in the setting budgeted training. Therefore, I don't see any necessity to spends two paragraphs in both introduction and related works on discussing core-set selections.

-- Flaw in the formula: For example, Eq (2) is wrong: If the prediction $ h^k_\theta $ is exactly the same as the gt $ y_i $, then $ p^t_i $ is being maximized but this is the easiest example (therefore $ p_i $ should have be minimized). Indeed, in Chang et. al. (2017), what they used was $ P_S(i \vert H, S_e, D) \propto 1 - \bar{p}(y_i \vert X_i) +\epsilon_D $. There is a negate in the front.

---

> ### Author Response · Authors · 2020-11-18
> **Response to Reviewer 3**
>
> We thank the reviewer for the valuable feedback and for the suggestions that we believe will improve the paper.
>
> #### **Clarify the main point:**
> We agree the title was misleading and will change it to "How Important is Importance Sampling for Deep Budgeted Training?" to address this (also raised by Rev. 1). The main motivation is to show that, to reduce the drop in accuracy when decreasing the training budget, it is better to introduce data augmentation rather than selecting specific samples. We will update the paper to clarify this.
>
> We will address the concerns regarding the introduction section (also raised by Rev. 4) and move the discussion about related work from the introduction to the related work and condense the text in both sections to transmit the main point of the paper more clearly.
>
> #### **Core-set selection:**
> We include core-set selection approaches to explore the potential of selecting the most important samples as an approach to reduce the budget of the training. We observe that it is more effective to randomly select a different subset of samples every iteration, which supports the main claim of the paper regarding the importance of variability in the data.
>
> #### **Additional comments:**
> We apologize for the typo in Eq. 2, we will correct the error. Note that the code used in all the experiments already includes the negation in the formula.

---

### Official Review · AnonReviewer1 · 2020-10-30
**well executed but narrow**

**Rating:** 5
**Confidence:** 4

**Review:**

This paper investigates the use of importance sampling in budgeted training.  Four importance sampling techniques from prior works are applied within the context of fixed training budgets, and compared under different conditions of training set selection, learning rate schedule and data augmentations.  Each aims to sample more useful examples more frequently, by using the loss or gradient magnitude as an importance measure.  Uniform sampling with and without replacement are used as baselines, and experiments are performed on cifar-10 and cifar-100.  The final conclusion is that importance sampling with budgets as low as 20% the original training schedule offer little if any improvement over uniform sampling, while additional data augmentations work well to make up lost validation accuracy.

While these are a thoroughly executed set of experiments on the cifar datasets, it's hard to know exactly what to make of negative results on just one dataset family.  Does importance sampling not work for budgeted training, or is cifar data not amenable to the technique?  Additional datasets or exploration of why it didn't work here would make this a lot stronger.

Similarly, I'm not sure exactly what to take away from the data augmentation experiment.  The conclusion states "in budgeted training ... data augmentation surpasses state-of-the-art importance sampling" --- but this seems to be the case in non-budgeted as well (SGD lines of table 3).  So is the conclusion that augmentation tends to work, for the training schedule lengths explored?

The budgets go down only to 20% of a 200-epoch training scheme (40 epochs), causing a baseline error increase from 5% to 8% in the worst case, still a fairly long schedule.  Evaluating smaller budgets, even down to near-chance performance level, might reveal areas with different behavior.

Overall, I find the experiments that were performed were well executed, and the negative result in this regime is a useful datapoint in assessing these methods.  Still, the experimental setups are a little too narrow (one dataset family, budgets starting at medium-length schedules) to draw any larger conclusions.


Additional comments:

- I feel the title is a little misleading, it implies that importance sampling is important, whereas the findings are the opposite.  Although maybe not technically wrong when read in hindsight (it merely raises the topic), I think it would make more sense if the title better reflected the findings, or was phrased as a question (eg "how important is ... ?"), which would leave a negative result more open

- related work section:  I think this it would also make sense to relate to few-shot learning, which is in a sense an extreme case of budget with just a handful of examples

- fig 1:  says scan-SGD but corresponding text in 4.4 says unif-SGD

---

> ### Author Response · Authors · 2020-11-18
> **Response for Reviewer 1**
>
> We thank the reviewer for the valuable feedback and for the suggestions that we believe will improve the paper.
>
> #### **Further exploration:**
> We agree with the reviewer in that more datasets would give a more robust picture (also raised by Rev. 2 and 4). Preliminary results in SVHN [1] show that the observations reported in the paper hold: with 30% of the budget and standard data augmentation (as in [2]) unif-SGD reaches 96.78 ± 0.13 of accuracy, p-SGD 96.77 ± 0.01, and SB 96.85 ± 0.01, whereas with RandAugment data augmentation 97.50 ± 0.07, 97.37 ± 0.10, and 97.43±0.19,  and with RICAP data augmentation 97.62 ± 0.16, 97.50 ± 0.05, and 97.40 ± 0.06. This confirms that importance sampling approaches (p-SGD and SB) bring little improvements over unif-SGD, compared to data augmentation techniques. In the final version of the paper, we will include the results for the different budgets and extend these experiments to mini-imagenet [3]. The following table summarizes these preliminary results:
>
> |30% budget | &nbsp;&nbsp;&nbsp;&nbsp;&nbsp; unif-SGD &nbsp;&nbsp;&nbsp;&nbsp;&nbsp;|&nbsp;&nbsp;&nbsp;&nbsp;&nbsp; p-SGD &nbsp;&nbsp;&nbsp;&nbsp;&nbsp;|&nbsp;&nbsp;&nbsp;&nbsp;&nbsp;&nbsp;&nbsp;&nbsp;&nbsp;   SB &nbsp;&nbsp;&nbsp;&nbsp;&nbsp;&nbsp;&nbsp;&nbsp;&nbsp;  |
> |:-----------|:-----------:|:-----------:|:-----------:|
> |Standard DA| 96.78 ± 0.13|96.77 ± 0.01|96.85 ± 0.01|
> |RandAugment| 97.50 ± 0.07|97.37 ± 0.10|**97.43±0.19**|
> |RICAP| **97.62 ± 0.16**|**97.50 ± 0.05**|97.40 ± 0.06|
>
> [1] Y. Netzer, T. Wang, A. Coates, A. Bissacco, Wu, B, and A. Ng, “Reading digits in natural images with unsupervised feature learning,” in Advances in Neural Information Processing Systems (NeurIPS), 2011.
>
> [2] S.-A. Rebuffi, S. Ehrhardt, K. Han, A. Vedaldi, and A. Zisserman, “SemiSupervised Learning with Scarce Annotations,” in IEEE International Conference on Computer Vision (ICCV), 2019.
>
> [3] O. Vinyals, C. Blundell, T. Lillicrap, K. Kavukcuoglu, and D. Wierstra, “Matching Networks for One Shot Learning,” in Advances in Neural Information Processing Systems (NeurIPS), 2016.
>
> #### **Clarify the main point:**
> The reviewer is correct: data augmentation does improve accuracy in budgeted and non-budgeted training. We will clarify the main point: given a limited budget, it is better to use data augmentation than to focus on selecting specific samples.
>
> #### **Budgets under 20%:**
> We experimented with extremely low budgets and reported some results at the end of Section 4. The main conclusion is that, in these extreme cases, stronger data augmentation techniques introduce too much variability in the data and the DNNs are unable to learn generalizable features in the given budget. We will add these results for CIFAR-10/100 in the appendix and will emphasize the limitations in extremely low budgets in Section 4.
>
> #### **Additional comments:**
> We will change the title to “How Important is Importance Sampling for Deep Budgeted Training?” and correct the error in Fig 1. The reviewer is correct in saying that few-shot learning shares similarity with training on an extremely small number of samples and we will point this out in the initial general paragraph of the related work. However, in that scenario the samples are given and cannot be changed during training, while we focus the related work on methods that train from scratch and, given a large set, assign importance or select a subset of samples.

---

### Author Response · Authors · 2020-11-23
**Changes in the paper**

We thank the reviewers for their suggestions and comments; they have been invaluable in improving the paper. The updated paper includes most of these suggestions, summarized below.

Changes to the text:
* Changed the paper title.
* Modified the abstract to make it more concise and address the main point more clearly.
* Clarified the explanations of data augmentation.
* Condensed the introduction and removed the discussion on related work.
* Included the suggested citations on few-shot learning, active learning, and curriculum learning.
* Rewrote the beginning of Section 3.
* Corrected the error in Equation 2.
* Added a note to stress that the row corresponding to SGD refers to the full training, not to one of the considered budgets.
* Highlighted the best results in the tables.
* Changed “early stopping” to “truncated training”.

Additional experiments added:
* Developed the explanation of extreme budgets (and included a new table in the appendix).
* Mentioned that the wall-clock time does not increase due to data augmentation (and included a table of wall-clock times in the appendix).
* Included experiments on SVHN and mini-ImageNet (due time constraints, we report only the 30% budget in mini-ImageNet but will extend this to the other budgets for the camera ready).

---

### Decision · Program_Chairs · 2021-01-07
**Final Decision**

**Decision:**

Reject

**Comment:**

This work investigates how importance sampling strategies can improve training with budgeted constraints., with a focus on the benefits from variety provided by data-augmentation samples.

Initial clarification issues raised by the reviewers were taken into account such as a new title, clarification of some explanations and corrections of typos.

However, the reviewers still agree that the paper is not ready for publication for several reasons:
- the comparison with the literature is still insufficient and should be better organised,
- experiments too narrow to conclude general benefits from the paper as it is, since there is a single type of tasks that is studied from the same dataset family. Questions related to very small budgets, below 20% also remain open and would require a new submission.